# Watching sports and happiness among older adults in Japan: The JAGES cohort study

**Kenjiro Kawaguchi**[1]*, **Kazushige Ide**[1], **Satoru Kanamori**[2,3], **Taishi Tsuji**[4], **Katsunori Kondo**[1,5]

1 Center for Preventive Medical Sciences, Chiba University, Chiba, Japan, 2 Teikyo University Graduate School of Public Health, Tokyo, Japan, 3 Department of Preventive Medicine and Public Health, Tokyo Medical University, Tokyo, Japan, 4 Institute of Health and Sport Sciences, University of Tsukuba, Tokyo, Japan, 5 Research Department, Institute for Health Economics and Policy, Tokyo, Japan

* kawaguchikenjiro@chiba-u.jp

## Abstract

### Objective

While watching sports may enhance older adults' happiness, the relationship between sports spectatorship and happiness may vary depending on on-site or TV/Internet. This study examined associations between different forms of sports spectatorship (on-site and TV/Internet) and happiness among older adults in Japan.

### Methods

We used data from the Japan Gerontological Evaluation Study, conducted in 2019 and 2022. The outcome variable was happiness, and the explanatory variables were watching sports on-site, watching professional sports on-site, and watching sports on TV/Internet. Multiple linear regression was performed to examine the association between watching sports and happiness, after adjusting for potential confounders. A modified Poisson regression analysis was performed for the binarized outcome with a cutoff of 8 points. Subgroup analyses were conducted based on sports club participation, age, and sex.

### Results

Watching sports on-site a few times annually was associated with higher happiness scores (B: 0.11, 95% CI: 0.03 to 0.19) and a higher prevalence of happiness (PR: 1.07, 95% CI: 1.03 to 1.12) than not watching sports. Similar results were observed when the participants watched professional sports a few times annually (B: 0.12, 95% CI: 0.02 to 0.22; PR: 1.06, 95% CI: 1.01 to 1.12). No significant associations were found between watching sports on TV/Internet and happiness. The associations were more pronounced among participants who did not participate in sports clubs, males, and those aged < 75 years.

### Conclusion

Watching sports on-site, particularly a few times a year, was associated with higher happiness levels among older adults. These findings highlight the importance of developing

**Data availability statement:** The data used in this study cannot be made publicly available due to ethical restrictions imposed by JAGES (the Japan Gerontological Evaluation Study) administrative office because the data contain potentially sensitive information about human research participants, including detailed health and socioeconomic status at small geographical areas. However, interested researchers may request access to the anonymized data by contacting the JAGES office: (Email: dataadmin.ml@jages.net; Website: https://www.jages.net/data_application/). Data requests will be considered following review of the research proposal and approval from the JAGES office. Researchers must agree to the JAGES data sharing guidelines and sign a data usage agreement to access the data. Data cannot be made public due to the inclusion of sensitive information provided by human participants. Researchers may request the data from the Japan Gerontological Evaluation Study (URL: https://www.jages.net/data_application/; Email: dataadmin.ml@jages.net).

**Funding:** This work was supported by JST RISTEX Japan Grant Number JPMJRS22B1. This study used data from JAGES (the Japan Gerontological Evaluation Study). The JAGES was supported by Grant-in-Aid for Scientific Research (20H00557, 20K10540, 21H03153, 21H03196, 21K17302, 22H00934, 22H03299, 22K04450, 22K13558, 22K17409, 23H00449, 23H03117), Health Labour Sciences Research Grants (19FA1012, 19FA2001, 21FA1012, 22FA2001, 22FA1010, 22FG2001), the Open Innovation Platform with Enterprises, Research Institute and Academia (JPMJOP1831) from the Japan Science and Technology (JST), a grant from Japan Health Promotion & Fitness Foundation, TMDU priority research areas grant and National Research Institute for Earth Science and Disaster Resilience. The funders had no role in study design, data collection and analysis, decision to publish, or preparation of the manuscript.

**Competing interests:** The authors have declared that no competing interests exist.

targeted interventions that promote older adults' access to live sports events as a public health strategy.

## Introduction

Promoting subjective well-being is crucial for healthy aging in later life [1]. Subjective well-being is defined as "subjective experiences and perceptions of individuals regarding their overall well-being and quality of life" [2]. Happiness, an essential aspect of subjective well-being, is characterized by a pleasant and slightly energized emotional state [3,4]. Research suggests that happiness can improve health outcomes; for instance, it can reduce mortality, lower morbidity, and increase functional independence [5]. However, maintaining good health is not the only benefit for happiness. Another critical reason for prioritizing happiness is its close connection with several other factors that play a significant role in leading a good life in later years. Several studies have reported that individuals with higher levels of subjective well-being, including happiness, have favorable outcomes in the personal, social, economic, and behavioral domains [6,7]. Given these findings, there is a growing interest in exploring factors and effective interventions that promote happiness and, more broadly, subjective well-being.

Being active later in life was positively associated with subjective well-being [3,8]. Numerous studies have found a positive association between physical activity, participation in active sports, and subjective well-being. One meta-analysis showed that physical activity positively affects the subjective well-being of older adults [9]. There is also a growing body of literature on the association between passive sports participation—such as watching sports on-site, on TV, or on the Internet (TV/Internet)—and happiness. Previous studies have reported that passive sports participation is associated with greater subjective well-being [10–12], including happiness [13]. Another interventional study involving older adults demonstrated that happiness was enhanced after observing a professional baseball game [14]. Attending sporting events can foster emotional support and a sense of belonging among spectators [15]. Spectator sports can create a common social identity that reflects various socioeconomic factors, demographics, and origins [16]. The experiences of belonging and identity are believed to contribute to increased happiness.

Passive sports participation may be a promising strategy to enhance happiness in older adults. Specifically, watching sports may be more accessible for older adults than engaging in sports or other physical activities. Additionally, accessibility to sports may vary depending on the form of spectatorship. Most individuals prefer watching sports on TV to live sports events because of their cost, emotional comfort, and convenience [17]. Previous research [10,18] has reported that roughly a quarter of Japanese older adults watch sports on-site at least once a year, and more than three-quarters watch sports on TV or the Internet at least once a year. According to a study focusing on college students [19], online video sport spectatorship may improve subjective well-being, with sport participation moderating the relationship between online video sport spectatorship and well-being. Thus, the relationship between sports spectatorship and happiness could differ depending on whether sports are watched on-site, on TV, or on the Internet. However, existing literature has not yet fully addressed this issue among older adults [20], which is particularly crucial in Japan, the most rapidly aging country worldwide. Therefore, using a nationwide sample, the current study investigated the association between watching sports on-site, on TV, or on the Internet and happiness among older adults in Japan. We hypothesized that older adults who engage in sports spectatorship would experience greater happiness than those who do not.

## Materials and methods

### Study design

Data were obtained from the Japan Gerontological Evaluation Study (JAGES) [21], an ongoing cohort study examining social factors influencing the health of community-dwelling individuals aged 65 years and older. Since 2003, JAGES surveys have been conducted approximately every three years in collaboration with municipal governments across Japan [21]. These governments develop and implement long-term care insurance plans, including preventive care services. JAGES surveys provide critical data for assessing policy effectiveness and guiding future planning [21]. Within each municipality, eligible individuals aged 65 years and older without long-term care certification are selected from the resident registry. In small municipalities, all eligible older adults are included, whereas in large municipalities, 30% of the target sample is randomly selected. Among the remaining 70%, individuals aged 65 to 67 years are chosen proportionally based on population distribution. Subsequently, individuals aged 68 years and older who participated in the most recent survey are prioritized for recruitment. Additional participants aged 68 years and older are enrolled until the target sample size is achieved.

We constructed an analytical sample for our study by combining data from the 2019 and 2022 waves. The survey questionnaires in both waves comprised core questions, and one of the eight modules was randomly assigned to the participants [21]. Each module contained a unique set of questions, with one explicitly asking about the form and frequency of watching sports during the past year. We obtained covariates from the 2019 wave, including the prior outcome of happiness and the outcome and explanatory variables from the 2022 wave. This design establishes a temporal relationship between the covariates and exposure, and ensures that the covariates precede exposure in time, avoiding adjustment for potential mediators [22]. Moreover, adjusting for prior outcomes can reduce the risk of reverse causation [22], which occurs when happy individuals tend to watch sports.

### Study population

The JAGES 2022 survey, conducted between November and December 2022, mailed questionnaires containing items related to sports spectatorship to 42,096 community-dwelling individuals aged ≥ 65 years, identified through official residential registers provided by municipal governments. Of them, 23,916 completed the questionnaire, resulting in a response rate of 56.8%. After excluding 2,395 respondents who lacked independence in activities of daily living (ADLs) or who had insufficient data on sex or age, the sample consisted of 21,521 eligible respondents. Lack of independence in ADLs was defined based on self-reported responses to the question, "Do you require nursing care or assistance in daily life from anyone?" [18]. Of them, 11,265 had completed the questionnaire in 2019 and were included in the final analysis.

### Measures

**Outcome variable: Happiness score.** In our research, we measured happiness levels using the following question: "On a scale of 0 to 10, where zero represents very unhappy and ten represents very happy, how would you rate your overall level of happiness?" Several studies have investigated the reliability and validity of single-item happiness scales with scores of 0–10, and the results have shown them to be reliable and valid measures of happiness [23,24].

**Explanatory variables: Frequency of watching sports.** We used the following questions to determine the frequency at which participants watched sports: 1) "On average, how often did you watch sports on-site in the past year? These include professional sports, local sports

clubs, and groups" (watching sports on-site); 2) "(For those who answered "yes" to the previous question) On average, how often did you watch professional sports on-site, such as at a stadium or arena, during the past year?" (watching professional sports on-site); 3) "On average, how often did you watch sports on television or the Internet in the past year? This includes not only professional sports but also local sports clubs and groups (excluding minimal viewing of television news)" (watching sports on TV/Internet). The participants were required to select one of the following response options: never, a few times a year, 1–3 times a month, or at least once a week. To ensure an adequate number of observations per category, we categorized each variable into three groups: never, a few times a year, and at least once a month.

**Covariates.** We included 23 covariates as possible confounders based on previous research [10,18] that examined the association between watching sports and depression or subjective well-being. We obtained all covariates detailed below from the 2019 wave. Sex (male or female), age (continuous), marital status (married or unmarried), living arrangement (living alone or with others), occupational status (employed or unemployed), years of education (< 10, 10–12, or ≥ 13 years), drinking status (none or current), and smoking status (none or current) were included in this study. The annual equivalent income, divided by the square root of the number of household members, was categorized as < 2 million, 2–4 million, or > 4 million JPY per year (1 USD = 155 JPY). Instrumental activities of daily living were assessed using the Tokyo Metropolitan Institute of Gerontology Index of Competence [25], which examines five activities: public transportation, shopping for daily necessities, preparing meals, paying bills, and handling banking. Each item was scored as 1 for yes (able to perform) or 0 for no, with participants classified as independent (5 points) or dependent (≤ 4 points). Self-reported height and weight were used to calculate the body mass index (kg/m$^2$). Self-rated health was evaluated using a standard one-item question: "How would you describe your current health?" The response categories (excellent, good, fair, and poor) were good (excellent or good) or poor (fair or poor). The Japanese short version of the Geriatric Depression Scale (GDS) was used to assess depressive symptoms. This 15-item scale, validated for older adults in Japan, has demonstrated good reliability (Cronbach's α = 0.80) and sensitivity (96%) and specificity (95%) for detecting major depressive disorder [26]. Scores range from 0 to 15, with higher scores indicating more severe depressive symptoms. Disease status included hypertension, stroke, cardiovascular disease, diabetes, dyslipidemia, and musculoskeletal disorders, which were assessed using yes/no responses. We included the participants' participation in hobbies and sports clubs. Participants were requested to indicate their frequency of participation by selecting one of the following options: "almost every day," "twice or thrice a week," "once a week," "once or twice a month," "a few times a year," or "never." Responses were classified as "yes" if individuals selected any of the five options ranging from "a few times a year" to "almost every day" and "no" if they chose "never." The population density per km$^2$ of the inhabitable area was categorized into three groups: < 1,000, 1,000–4,000, and > 4,000 persons/km$^2$. The happiness in the 2019 wave was included as a covariate.

## Statistical analysis

Multiple linear regression analyses were performed to examine the association between sports participation and happiness. We created separate regression models for each form of watching sports on-site, professional sports on-site, and sports on TV or the Internet. We individually entered each of the three explanatory variables into the statistical models. Additionally, given the overlap between those who watch sports on TV/Internet and those who watch sports on-site, we constructed regression models that encompassed watching sports on-site and on TV/Internet, as well as watching professional sports on-site and watching sports on TV/

Internet simultaneously. The non-spectator category was used as the reference for each form. Unstandardized coefficients (B), standardized coefficients (β) and 95% confidence intervals (CIs) were calculated. To enhance the interpretation of our findings, we dichotomized the outcome variable of happiness. We determined cutoff points of 7 and 8 based on the average scale of the outcome variable. We estimated the prevalence ratio (PR) using a modified Poisson regression method because the prevalence of dichotomized outcomes was greater than 10% [27]. Furthermore, we conducted subgroup analyses to investigate potential differences in the associations between watching sports and happiness stratified by sports club participation (yes or no), sex, and age (< 75 years or ≥ 75 years) in the 2019 wave. These factors could influence accessibility to sports due to differences in interest, engagement, mobility, health concerns, and gender-related preferences. The subgroup analyses aimed to understand whether the effects of watching sports on happiness were equally distributed, or whether certain subgroups benefited more from accessing or engaging in watching sports. To assess multicollinearity, we calculated the Variance Inflation Factor (VIF) for each model. All VIFs values were less than 10, indicating no significant multicollinearity in the models. Missing values were imputed using random forest imputation, an iterative method based on random forests that constitutes multiple imputation schemes, by averaging many unpruned classification or regression trees [28]. All statistical analyses were performed using R version 4.2.1, with p < 0.05 indicating statistical significance.

This study was approved by the Ethics Committee of Chiba University, Japan (approval number: M10460). All participants were informed that their participation in the study would be voluntary. After completing the questionnaire, selecting the acceptance checkbox, and returning it via mail, the participants provided written consent to participate.

## Results

Table 1 summarizes the background characteristics of the 11,265 participants before imputation for missing values. The average age was 73.6 years (standard deviation [SD], 5.6) and the proportion of women was 52.2% (n = 5,883). Of the 11,265 participants, 418 (4.1%) reported watching sports on-site at least once a month, whereas 1,222 (12.0%) and 8,511 (83.8%) reported watching sports on-site a few times a year or never, respectively. Similarly, 64 (0.6%) participants reported watching professional sports on-site at least once a month, whereas 625 (6.2%) and 9,329 (93.1%) reported watching professional sports on-site a few times a year or never, respectively. Of the participants, 6,113 (58.8%) reported watching sports on TV or the Internet at least once a month. A total of 2,434 (23.4%) participants reported watching sports on TV or the Internet a few times a year, whereas 1,857 (17.8%) had never watched sports on TV or the Internet. The average happiness score for the 2022 wave is 7.4 (SD 1.8). S1 Table in S1 File shows the distribution of happiness scores by the frequency of watching sports. Participants who watched sports on-site a few times a year reported higher happiness scores (mean = 7.6) than those who did not (mean = 7.2). Regarding professional sports on-site, happiness scores were similar between those who never watched (mean = 7.6) and those who watched it a few times a year (mean = 7.7). Participants who watched sports on TV/Internet at least once a month showed higher happiness scores (mean = 7.4) than those who did not (mean = 7.1). To assess potential selection bias from sample attrition, we compared our analytic sample with the original sample in 2019 (n = 22,528; S2 Table in S1 File). The background characteristics of our analytic sample were largely comparable to those of the original sample, although our analytic samples included slightly younger participants (mean age: 73.6 vs 74.5 years).

Table 2 shows the results of the linear regression analysis of watching sports and happiness among independent older adults aged ≥ 65 years. After adjusting for possible confounders, participants who watched sports on-site a few times a year had higher happiness scores than

**Table 1. Characteristics of the participants aged 65 or older before missing values were imputed.**

| Characteristic | N = 11,265[a] |
|---|---|
| Sex, female | 5,883 (52.2%) |
| Age (years) | 73.6 (5.6) |
| Marital status: married | 8,036 (72.2%) |
| Living arrangements: living alone | 1,455 (13.6%) |
| Occupational status: employed | 3,397 (32.8%) |
| Years of education (years) | |
| <10 | 2,559 (23.3%) |
| 10–12 | 4,908 (44.7%) |
| ≥ 13 | 3,523 (32.1%) |
| Drinking status: yes | 4,766 (43.8%) |
| Smoking status: yes | 1,059 (9.6%) |
| Equivalent income (yen) | |
| < 2 million | 4,818 (48.7%) |
| 2–4 million | 4,034 (40.8%) |
| > 4 million | 1,041 (10.5%) |
| Body mass index (kg/m$^2$) | 23.0 (3.1) |
| Instrumental activities of daily living, independent | 10,236 (93.2%) |
| Self-rated health, good | 10,076 (91.5%) |
| Geriatric depression scale (0–15: higher scores indicate a more severe presence of depressive symptoms) | 2.7 (2.8) |
| Hypertension, yes | 5,179 (47.3%) |
| Stroke, yes | 245 (2.2%) |
| Cardiovascular disease, yes | 1,025 (9.4%) |
| Diabetes, yes | 1,645 (15.0%) |
| Dyslipidemia, yes | 1,763 (16.1%) |
| Musculoskeletal disorders, yes | 1,048 (9.6%) |
| Population density (persons/km$^2$) | |
| < 1,000 | 4,233 (37.6%) |
| 1,000–4,000 | 3,305 (29.3%) |
| > 4,000 | 3,727 (33.1%) |
| Participation in sports clubs, yes | 3,543 (36.4%) |
| Participation in hobby groups, yes | 4,493 (45.4%) |
| Happy score in 2019 (0～10: higher scores indicate feeling happier) | 7.3 (1.8) |
| Happy score in 2022 (0～10: higher scores indicate feeling happier) | 7.4 (1.8) |
| Watching sports on-site | |
| Never | 8,511 (83.8%) |
| A few times a year | 1,222 (12.0%) |
| At least once a month | 418 (4.1%) |
| Watching professional sports on-site | |
| Never | 9,329 (93.1%) |
| A few times a year | 625 (6.2%) |
| At least once a month | 64 (0.6%) |
| Watching sports on TV/Internet | |
| Never | 1,857 (17.8%) |
| A few times a year | 2,434 (23.4%) |
| At least once a month | 6,113 (58.8%) |

[a]n (%); Mean (Standard deviation).

**Table 2. Regression coefficients and 95% confidence intervals of linear regression analyses examining associations between happiness and watching sports.**

| Frequency of watching sports | B (95% CI) | β (95% CI) | p-value |
|---|---|---|---|
| **Watching sports on-site** | | | |
| Never | Reference | Reference | |
| A few times a year | 0.11 (0.03, 0.19) | 0.06 (0.02, 0.11) | 0.006 |
| At least once a month | 0.07 (−0.05, 0.20) | 0.04 (−0.03, 0.11) | 0.262 |
| **Watching professional sports on-site** | | | |
| Never | Reference | Reference | |
| A few times a year | 0.12 (0.02, 0.22) | 0.07 (0.01, 0.13) | 0.018 |
| At least once a month | 0.01 (−0.29, 0.31) | 0.00 (−0.16, 0.17) | 0.952 |
| **Watching sports on TV/Internet** | | | |
| Never | Reference | Reference | |
| A few times a year | −0.04 (−0.11, 0.04) | −0.02 (−0.06, 0.02) | 0.357 |
| At least once a month | 0.03 (−0.04, 0.10) | 0.02 (−0.02, 0.06) | 0.337 |

Note: All models were controlled for sex, age, marital status, living arrangements, occupational status, years of education, drinking status, smoking status, equivalized income, body mass index, instrumental activities of daily living, self-rated health, geriatric depression scale, hypertension, stroke, cardiovascular disease, diabetes, dyslipidemia, musculoskeletal disorders, population density, participation in sports clubs, participation in hobbies, and happiness scores in 2019. B: Unstandardized coefficient, β: Standardized coefficient, CI: Confidence interval.

those who did not (B: 0.11, 95% CI: 0.03 to 0.19; β: 0.06, 95% CI: 0.02 to 0.11). However, watching sports on-site at least once a month was not significantly associated with happiness (B: 0.07, 95% CI: −0.05 to 0.20; β: 0.04, 95% CI: −0.03 to 0.11). Concerning watching professional sports on-site, watching a few times a year was significantly associated with a higher happiness score (B: 0.12, 95% CI: 0.02 to 0.22; β: 0.07, 95% CI: 0.01 to 0.13), but watching at least once a month was not (B: 0.01, 95% CI: −0.29 to 0.31; β: 0.00, 95% CI: −0.16 to 0.17). In terms of watching sports on TV or the Internet, no significant associations were observed between watching sports and happiness, regardless of the frequency of watching (a few times a year B: −0.04, 95% CI: −0.11 to 0.04;; β: −0.02, 95% CI: −0.06 to 0.02; at least once a month B: 0.03, 95% CI: −0.04 to 0.10; β: 0.02, 95% CI: −0.02 to 0.06). Table 3 presents the regression model results. When holding covariates constant, on average, individuals who watched sports on-site a few times a year reported 0.11 points higher happiness than those who did not. This increase in happiness is comparable to an increase in income from JPY 2 million to JPY 4 million. Similar results were obtained in the analyses in which two explanatory variables–watching sports on TV or the Internet and watching sports or professional sports on-site–were considered simultaneously (S3 and S4 Tables in S1 File).

The findings of the modified Poisson regression analysis of the relationship between watching sports and happiness are presented in Table 4. After adjusting for potential confounders, participants who watched sports on-site a few times a year and those who watched sports on-site at least once a month had a higher prevalence of happiness (happiness score ≥ 8) than those who did not (PR: 1.07, 95% CI: 1.03 to 1.12 and PR: 1.07, 95% CI: 1.00 to 1.14, respectively). Participants who watched professional sports on-site a few times a year, not at least once a month, were more likely to report feeling happy than those who did not (PR: 1.06, 95% CI: 1.01 to 1.12; PR: 1.06, 95% CI: 0.89 to 1.26, respectively). Similar results were obtained in the analyses that considered two explanatory variables (S5 and S6 Tables in S1 File).

In the subgroup analysis of participation in sports clubs at least a few times a year (n = 4,091), none of the three forms of watching sports was associated with a higher happiness score (as presented in Table 5). However, the subgroup analysis of non-participation in sports clubs (n = 7,174) observed that individuals who watched sports on-site a few times a year and

**Table 3. Regression coefficients for all covariates included in the models with each explanatory variable of watching sports on-site, watching professional sports on-site, and watching sports on TV/Internet.**

| Explanatory Variables and Covariates: Variable (Reference Group) | B | p-value | B | p-value | B | p-value |
|---|---|---|---|---|---|---|
| Watching sports on-site (Never) | | | | | | |
| A few times a year | 0.11 | 0.006 | | | | |
| At least once a month | 0.07 | 0.262 | | | | |
| Watching professional sports on-site (Never) | | | | | | |
| A few times a year | | | 0.12 | 0.018 | | |
| At least once a month | | | 0.01 | 0.952 | | |
| Watching sports on TV/Internet (Never) | | | | | | |
| A few times a year | | | | | −0.04 | 0.357 |
| At least once a month | | | | | 0.03 | 0.337 |
| Sex, male (female) | −0.21 | <0.001 | −0.21 | <0.001 | −0.21 | <0.001 |
| Age | 0.01 | <0.001 | 0.01 | <0.001 | 0.01 | <0.001 |
| Marital status, married (unmarried) | 0.01 | 0.856 | 0.01 | 0.875 | 0.00 | 0.930 |
| Living arrangements, living alone (living with others) | −0.08 | 0.087 | −0.08 | 0.083 | −0.08 | 0.084 |
| Occupational status, employed (unemployed) | −0.00 | 0.939 | −0.00 | 0.970 | 0.00 | 0.948 |
| Years of education (< 10 years) | | | | | | |
| 10–12 years | −0.00 | 0.943 | −0.00 | 0.946 | −0.00 | 0.898 |
| ≥ 13 years | 0.10 | 0.006 | 0.10 | 0.006 | 0.10 | 0.009 |
| Drinking status, current (none) | 0.02 | 0.519 | 0.02 | 0.507 | 0.02 | 0.511 |
| Smoking status, current (none) | −0.06 | 0.189 | −0.06 | 0.183 | −0.06 | 0.183 |
| Equivalent income (< 2 million yen) | | | | | | |
| 2–4 million yen | 0.07 | 0.017 | 0.07 | 0.017 | 0.06 | 0.020 |
| > 4 million yen | 0.11 | 0.013 | 0.11 | 0.014 | 0.11 | 0.013 |
| Body mass index (kg/m$^2$) | 0.00 | 0.879 | 0.00 | 0.872 | 0.00 | 0.860 |
| Instrumental activities of daily living, independent (dependent) | −0.09 | 0.085 | −0.09 | 0.089 | −0.09 | 0.086 |
| Self-rated health, good (poor) | −0.01 | 0.901 | −0.01 | 0.884 | −0.01 | 0.914 |
| Participation in sports clubs, yes (no) | −0.03 | 0.246 | −0.03 | 0.337 | −0.03 | 0.376 |
| Participation in hobby groups, yes (no) | 0.03 | 0.377 | 0.03 | 0.347 | 0.03 | 0.327 |
| Happy score in 2019 | 0.53 | <0.001 | 0.53 | <0.001 | 0.53 | <0.001 |
| Geriatric depression scale | −0.10 | <0.001 | −0.10 | <0.001 | −0.10 | <0.001 |
| Hypertension, yes (no) | −0.03 | 0.306 | −0.03 | 0.279 | −0.03 | 0.298 |
| Stroke, yes (no) | −0.09 | 0.292 | −0.08 | 0.308 | −0.09 | 0.293 |
| Cardiovascular disease, yes (no) | −0.04 | 0.394 | −0.04 | 0.384 | −0.04 | 0.390 |
| Diabetes, yes (no) | 0.03 | 0.426 | 0.03 | 0.412 | 0.03 | 0.396 |
| Dyslipidemia, yes (no) | −0.00 | 0.899 | −0.00 | 0.902 | −0.00 | 0.886 |
| Musculoskeletal disorders, yes (no) | −0.10 | 0.019 | −0.10 | 0.018 | −0.10 | 0.018 |
| Population density (< 1,000 persons/km$^2$) | | | | | | |
| 1,000–4,000/km$^2$ | −0.02 | 0.544 | −0.02 | 0.491 | −0.02 | 0.477 |
| >4,000/km$^2$ | −0.06 | 0.038 | −0.07 | 0.026 | −0.07 | 0.030 |

those who watched sports on-site at least once a month, as well as those who watched professional sports on-site a few times a year, had a higher happiness score than those who did not (B: 0.17, 95% CI: 0.06 to 0.28; β: 0.09, 95% CI: 0.03 to 0.16; B: 0.29, 95% CI: 0.09 to 0.50; β: 0.16, 95% CI: 0.05 to 0.27 and B: 0.17, 95% CI: 0.02 to 0.33; β: 0.09, 95% CI: 0.01 to 0.18, respectively; Table 5). In the sex-stratified analysis (male: n = 5,382; female: n = 5,883), men who watched sports on-site a few times a year and professional sports on-site a few times a

**Table 4. Prevalence ratios and 95% confidence intervals of modified Poisson regression analyses examining the associations between happiness and watching sports.**

| Frequency of watching sports | PR | 95% CI | p-value |
|---|---|---|---|
| **Watching sports on-site** | | | |
| Never | Reference | | |
| A few times a year | 1.07 | 1.03, 1.12 | 0.001 |
| At least once a month | 1.07 | 1.00, 1.14 | 0.037 |
| **Watching professional sports on-site** | | | |
| Never | Reference | | |
| A few times a year | 1.06 | 1.01, 1.12 | 0.027 |
| At least once a month | 1.06 | 0.89, 1.26 | 0.503 |
| **Watching sports on TV/Internet** | | | |
| Never | Reference | | |
| A few times a year | 0.98 | 0.93, 1.03 | 0.358 |
| At least once a month | 1.01 | 0.96, 1.05 | 0.814 |

Note: All models controlled for sex, age, marital status, living arrangements, occupational status, years of education, drinking status, smoking status, equivalized income, body mass index, instrumental activities of daily living, self-rated health, geriatric depression scale, hypertension, stroke, cardiovascular disease, diabetes, dyslipidemia, musculoskeletal disorders, population density, participation in sports clubs, participation in hobbies, and happiness scores in 2019. CI: Confidence interval, PR: Prevalence ratio.

year were more likely to feel happier than those who did not (B: 0.16, 95% CI: 0.06 to 0.26; β: 0.09, 95% CI: 0.04 to 0.15; B: 0.20, 95% CI: 0.08 to 0.32; β: 0.11, 95% CI: 0.04 to 0.18, respectively), whereas no significant associations were observed among women (S6 Table in S1 File). In the age-stratified analysis (< 75 years: n = 6,653; ≥ 75 years: n = 4,612), individuals aged <75 years who watched sports on-site or professional sports on-site a few times a year were more likely to report higher levels of happiness than those who did not (B: 0.10, 95% CI: 0.00 to 0.20; β: 0.06, 95% CI: 0.00 to 0.11; B: 0.14, 95% CI: 0.01 to 0.27; β: 0.08, 95% CI: 0.01 to 0.15, respectively, as shown in S8 Table in S1 File). Conversely, individuals aged 75 years or older who watched sports on TV/Internet a few times a year were more likely to report lower levels of happiness than those who did not (B: − 0.15, 95% CI: − 0.28 to − 0.02; β: − 0.08, 95% CI: − 0.15 to − 0.01, as shown in S8 Table in S1 File).

## Discussion

To the best of our knowledge, this is the first study to focus on the differences in the relationship between watching sports and happiness based on spectating type. These findings revealed that watching sports on-site, particularly a few times a year, was associated with higher happiness levels among older adults. Conversely, the relationship between watching sports on TV/Internet was insignificant. This study confirmed a positive association between watching sports and happiness, with robust associations observed among individuals who did not participate in sports clubs (i.e., those with fewer opportunities to engage in sports), men, and those under 75 years of age.

Consistent with our findings, studies on watching sports have shown a positive association with subjective well-being [14,15,29–31]. Factors such as team identification, game outcomes, and game processes can mediate the relationship between watching sports on-site and happiness [31–33]. In terms of biological mechanisms, watching popular sports may increase brain activity and structural volume in regions associated with positive emotions, leading to enhanced self-reported well-being [34]. We observed associations between watching sports on-site and happiness, regardless of whether it was live spectatorship of professional sports.

**Table 5. Regression coefficients and 95% confidence intervals of linear regression analyses examining the associations between happiness and watching sports stratified by participation in sports clubs.**

| Frequency of watching sports | B (95% CI) | β (95% CI) | p-value |
|---|---|---|---|
| **Participation in sports clubs (n = 4,091)** | | | |
| **Watching sports on-site** | | | |
| Never | Reference | Reference | |
| A few times a year | 0.04 (−0.06, 0.15) | 0.03 (−0.04, 0.09) | 0.413 |
| At least once a month | −0.08 (−0.24, 0.07) | −0.05 (−0.14, 0.04) | 0.281 |
| **Watching professional sports on-site** | | | |
| Never | Reference | Reference | |
| A few times a year | 0.07 (−0.06, 0.21) | 0.04 (−0.04, 0.12) | 0.295 |
| At least once a month | −0.36 (−0.76, 0.04) | −0.21 (−0.46, 0.03) | 0.078 |
| **Watching sports on TV/Internet** | | | |
| Never | Reference | Reference | |
| A few times a year | 0.02 (−0.11, 0.16) | 0.02 (−0.07, 0.10) | 0.723 |
| At least once a month | 0.10 (−0.02, 0.23) | 0.06 (−0.01, 0.14) | 0.105 |
| **No participation in sports clubs (n = 7,174)** | | | |
| **Watching sports on-site** | | | |
| Never | Reference | Reference | |
| A few times a year | 0.17 (0.06, 0.28) | 0.09 (0.03, 0.16) | 0.003 |
| At least once a month | 0.29 (0.09, 0.50) | 0.16 (0.05, 0.27) | 0.005 |
| **Watching professional sports on-site** | | | |
| Never | Reference | Reference | |
| A few times a year | 0.17 (0.02, 0.33) | 0.09 (0.01, 0.18) | 0.029 |
| At least once a month | 0.38 (−0.05, 0.81) | 0.21 (−0.03, 0.44) | 0.085 |
| **Watching sports on TV/Internet** | | | |
| Never | Reference | | |
| A few times a year | −0.06 (−0.15, 0.04) | −0.03 (−0.08, 0.02) | 0.219 |
| At least once a month | 0.01 (−0.08, 0.09) | 0.00 (−0.04, 0.05) | 0.906 |

Note: All models controlled for sex, age, marital status, living arrangements, occupational status, years of education, drinking status, smoking status, equivalized income, body mass index, instrumental activities of daily living, self-rated health, geriatric depression scale, hypertension, stroke, cardiovascular disease, diabetes, dyslipidemia, musculoskeletal disorders, population density, participation in sports clubs, participation in hobbies, and happiness scores in 2019. B: Unstandardized coefficient, CI: Confidence interval.

Based on our findings, watching sports, irrespective of the competition level, can elicit feelings of happiness because it provides an opportunity to support one's favorite teams, athletes, friends, and family [18].

Our findings suggest that individuals who watch sports on-site more frequently are not necessarily happy. While frequent sports watching might intuitively seem more beneficial, a moderate frequency (a few times per year) appears optimal for happiness. The complex relationship between watching frequency and happiness can be explained through three mechanisms. First, regular spectators often experience stronger negative emotions from losses, which can offset positive effects [35]. Moderate attendance may provide a balance by allowing individuals to enjoy the excitement of live events while ensuring sufficient emotional recovery between experiences. This pattern can help minimize the accumulation of negative emotions resulting from frequent exposure to team losses or disappointments. Second, quality of engagement appears to be more critical for happiness than quantity of attendance. Evidence on the connection between watching sports and well-being [36] indicates that an individual's

psychological connection with a sports team may be more important than watching frequency itself for greater well-being. The depth of emotional attachment to a favorite team, rather than the frequency of attendance, may drive the benefits of happiness associated with sports spectatorship. Third, hedonic adaptation theory offers additional insights into why occasional attendance may yield greater happiness benefits. This theory suggests that people tend to return to a baseline level of happiness despite repeated positive experiences [37]. While occasional attendance may provide novelty, excitement, and a temporary boost in mood, frequent attendance may reduce positive emotional returns as individuals become accustomed to the experience. Less frequent attendance also enhances the psychological benefits of anticipation and savoring, further amplifying the positive emotional effects of each event. Moderate attendance may balance excitement with emotional recovery, avoiding negative emotions associated with frequent losses, or team-related disappointments. Thus, the relationship between watching sports and happiness appears to be influenced by both the quantitative and qualitative aspects of sports engagement.

The positive association between watching sports and happiness was specific to on-site spectatorship and not observed with TV/Internet watching. One study found that viewing sporting events on TV or the Internet could enhance viewers' well-being [38]. Conversely, another study examining the effects of the frequency of watching professional baseball on fans' life satisfaction revealed that while attending ballparks to watch games was positively associated with sports fans' life satisfaction, watching baseball on the TV, Internet, or mobile devices was not [39]. While watching sports on TV or the Internet is routine, visiting a stadium or arena for spectatorship is viewed as a special event because it requires more time and money. Additionally, attending a sporting event provides an opportunity to escape from daily life, appreciate athletes' aesthetics, and socialize with friends or family, which can make the experience more immersive [40]. Thus, it is reasonable to assume that watching sports on-site can significantly affect happiness more than watching sports on TV or the Internet.

The relationship between watching sports and happiness was observed among older adults who did not participate in sports clubs rather than among those who participated at least several times a year. The effect of watching sports on happiness may vary depending on the active participation of older adults in sports. Many studies have consistently demonstrated that active participation in sports can lead to higher happiness levels [9]. However, it is generally difficult to encourage older adults, especially those with functional decline or limited mobility, to engage in regular sports or exercise. Our findings suggest that watching sports may have different implications for participants and non-participants in sports clubs. Sports club participants may already receive social and emotional benefits through active participation, creating a ceiling effect for additional benefits from watching sports. By contrast, non-participants may experience unique social connections and sports-related enjoyment, primarily through watching sports. Additionally, the novelty effect may be stronger for non-participants. While sports club participants are regularly exposed to sports environments, individuals who do not participate in sports may find the atmosphere and excitement of live sports events more stimulating and beneficial to their emotional states. However, these potential mechanisms remain hypothetical, and future research is needed to examine the psychological processes underlying the differential effects of watching sports between sports participants and non-participants.

Previous research has reported sex-specific differences in the enjoyment of watching sporting events on TV [41]. Men were more likely to engage in the passionate and emotional aspects of watching sports on TV, be immersed in the excitement and drama of the game, and actively participate in the watching experience by eating, drinking, and vocalizing their reactions. In contrast, women were more likely to watch sports to pass time or companionship. Other studies have demonstrated that male viewers prefer sports featuring physical

competition or the use of tools such as boxing, ice hockey, and basketball, whereas female viewers exhibit a greater affinity for aesthetic sports such as figure skating [42]. The relationship between watching sports and happiness may vary depending on how men and women watch sports.

We observed no notable association between watching sports and happiness in individuals aged ≥ 75 years, contrary to the findings for those aged < 75 years. Older adults tend to choose media content that makes them feel good and avoid programs that are negative in nature [43]. As individuals age, they often prioritize experiences that are more emotionally meaningful [44], and their emotional stability tends to increase [45]. This suggests that those aged ≥ 75 years who do not watch sports may have alternative ways of increasing their happiness apart from watching sports. However, this proposed explanation cannot fully account for our finding that those aged ≥ 75 years who watched sports on TV or the Internet a few times per year were less happy than those who did not. However, further studies are required to address this issue.

Our research revealed that on-site sports spectatorship, but not TV or Internet viewing, is positively associated with happiness among older adults. While watching sports on TV or the Internet is accessible, visiting stadiums or arenas for spectatorship may be more effective in enhancing happiness [40]. Facilitating access to live sports events in older adults may function as a public health strategy to enhance happiness. This strategy could include both financial interventions, such as targeted ticket discounts or voucher programs [18], and infrastructure optimization of existing facilities, such as public stadiums and parks [46]. Stadiums and arenas could integrate age-friendly features, including ergonomic seating, accessible restrooms, and designated rest areas, to better accommodate older spectators. Hosting pre-event social gatherings or arranging guided group visits might also improve attendance and foster greater engagement among older adults. Such comprehensive approaches to promoting live sports spectatorship may serve as an effective public health strategy to enhance happiness among older adults.

One strength of this study is that it examined the relationship between watching sports and happiness in older adults using a large population-based sample. Additionally, we used the covariates measured before measuring the exposure variables of interest to reduce the possibility of reverse causation. In other words, we aimed to account for the possibility that happier older adults were more inclined to watch sports.

This study has several limitations. First, generalizability is limited by our focus on community-dwelling older adults in Japan. Cultural factors may influence the relationship between sports spectatorship and happiness, warranting caution when extending these findings to other populations. Second, while our analytic sample was largely comparable to the original sample, the sample attrition could have introduced selection bias, potentially skewing the sample towards more active individuals. Additionally, although we adjusted for several confounders, unmeasured variables such as sporting event type, motivation for attendance, and team preferences were not considered due to questionnaire constraints [47]. The COVID-19 pandemic may also have affected our data, as concerns about infection could have diminished enjoyment and attenuated feelings of happiness among spectators [48]. Furthermore, the primarily cross-sectional design restricts our ability to infer causality. Although we attempted to reduce reverse causation by using covariates measured prior to exposure, we cannot eliminate the possibility that happier individuals are simply more inclined to watch sports. Future research should address these limitations by replicating this study in diverse populations across different cultures. Longitudinal designs are needed to establish temporal relationships and elucidate the dynamic interplay between sports spectatorship and happiness. Further investigation into underlying mechanisms (e.g., social

interaction, emotional responses, team identification) and the moderating effects of sport type and viewing context is also warranted.

## Conclusion

In conclusion, this study highlights the positive association between watching sports on-site and happiness among older adults. However, watching sports on TV or the Internet was not associated with higher levels of happiness. Additionally, the association between watching sports and happiness was pronounced among individuals who did not participate in sports, men, and those aged < 75 years. Interventions aimed at promoting access to live sporting events could significantly increase happiness.

## Supporting information

**S1 File.   S1 Table.** Distribution of happiness scores by frequency of watching sports. **S2 Table.** Comparison of background characteristics between analytic sample and original sample in 2019. **S3 Table.** Regression coefficients and 95% confidence intervals of linear regression analyses examining the associations between happiness and watching sports in the model, including the two explanatory variables of watching sports on-site and on TV/Internet simultaneously. **S4 Table.** Regression coefficients and 95% confidence intervals of linear regression analyses examining the associations between happiness and watching sports in the model, including the two explanatory variables of watching professional sports on-site and watching sports on TV/Internet simultaneously. **S5 Table.** Prevalence ratios and 95% confidence intervals of modified Poisson regression analyses examining the associations between happiness and watching sports in the model, including the two explanatory variables of watching sports on-site and on TV/Internet simultaneously. **S6 Table.** Prevalence ratios and 95% confidence intervals of modified Poisson regression analyses examining the associations between happiness and watching sports in the model, including the two explanatory variables of watching professional sports on-site and watching sports on TV/Internet simultaneously. **S7 Table.** Regression coefficients and 95% confidence intervals of linear regression analyses examining the association between happiness and watching sports stratified by sex. **S8 Table.** Regression coefficients and 95% confidence intervals of linear regression analyses examining the associations between happiness and watching sports stratified by age (< 75 years and ≥ 75 years). (DOCX)

## Acknowledgments

We would like to thank Editage [http://www.editage.com] for editing and reviewing this manuscript for English language.

## Author contributions

**Conceptualization:** Kenjiro Kawaguchi.

**Formal analysis:** Kenjiro Kawaguchi.

**Investigation:** Kenjiro Kawaguchi.

**Methodology:** Kenjiro Kawaguchi.

**Supervision:** Kazushige Ide, Satoru Kanamori, Taishi Tsuji, Katsunori Kondo.

**Writing – original draft:** Kenjiro Kawaguchi.

**Writing – review & editing:** Kenjiro Kawaguchi, Kazushige Ide, Satoru Kanamori, Taishi Tsuji, Katsunori Kondo.

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
