## [Decision Letter · Decision Letter 0]

22 Oct 2024

PONE-D-24-42023Watching sports and happiness among older adults in Japan: The JAGES cohort studyPLOS ONE

Dear Dr. Kawaguchi,

Thank you for submitting your manuscript to PLOS ONE. After careful consideration, we feel that it has merit but does not fully meet PLOS ONE’s publication criteria as it currently stands. The reviewers have highlighted several minor issues that need to be addressed before the manuscript can be considered for final acceptance.

We encourage you to revise your manuscript in response to the reviewers' comments, ensuring that all points raised are thoroughly addressed. Once revised, please submit your updated manuscript for further consideration.

If applicable, we recommend that you deposit your laboratory protocols in protocols.io to enhance the reproducibility of your results. Protocols.io assigns your protocol its own identifier (DOI) so that it can be cited independently in the future. For instructions, see: https://journals.plos.org/plosone/s/submission-guidelines#loc-laboratory-protocols . Additionally, PLOS ONE offers an option for publishing peer-reviewed Lab Protocol articles, which describe protocols hosted on protocols.io. Read more information on sharing protocols at https://plos.org/protocols?utm_medium=editorial-email&utm_source=authorletters&utm_campaign=protocols .

We look forward to receiving your revised manuscript.

Kind regards,

Mohamed Ahmed Said, Ph.D.

Academic Editor

PLOS ONE

“This work was supported by JST RISTEX Japan Grant Number JPMJRS22B1. This study used data from JAGES (the Japan Gerontological Evaluation Study). The JAGES was supported by Grant-in-Aid for Scientific Research (20H00557, 20K10540, 21H03153, 21H03196, 21K17302, 22H00934, 22H03299, 22K04450, 22K13558, 22K17409, 23H00449, 23H03117), Health Labour Sciences Research Grants (19FA1012, 19FA2001, 21FA1012, 22FA2001, 22FA1010, 22FG2001), the Open Innovation Platform with Enterprises, Research Institute and Academia (JPMJOP1831) from the Japan Science and Technology (JST), a grant from Japan Health Promotion & Fitness Foundation, TMDU priority research areas grant and National Research Institute for Earth Science and Disaster Resilience.”

“We would like to thank Editage [http://www.editage.com] for editing and reviewing this manuscript for English language. This study was supported by JST RISTEX, Japan (grant number JPMJRS22B1). This study used data from the JAGES. The JAGES was supported by Grant-in-Aid for Scientific Research (20H00557, 20K10540, 21H03153, 21H03196, 21K17302, 22H00934, 22H03299, 22K04450, 22K13558, 22K17409, 23H00449, and 23H03117), Health Labour Sciences Research Grants (19FA1012, 19FA2001, 21FA1012, 22FA2001, 22FA1010, and 22FG2001), the Open Innovation Platform with Enterprises, Research Institute and Academia (JPMJOP1831) from the Japan Science and Technology (JST), a grant from Japan Health Promotion & Fitness Foundation, TMDU priority research areas grant, and National Research Institute for Earth Science and Disaster Resilience. The views and opinions expressed in this article are those of the authors and do not necessarily reflect the official policies or positions of the respective funding organizations.”

“This work was supported by JST RISTEX Japan Grant Number JPMJRS22B1. This study used data from JAGES (the Japan Gerontological Evaluation Study). The JAGES was supported by Grant-in-Aid for Scientific Research (20H00557, 20K10540, 21H03153, 21H03196, 21K17302, 22H00934, 22H03299, 22K04450, 22K13558, 22K17409, 23H00449, 23H03117), Health Labour Sciences Research Grants (19FA1012, 19FA2001, 21FA1012, 22FA2001, 22FA1010, 22FG2001), the Open Innovation Platform with Enterprises, Research Institute and Academia (JPMJOP1831) from the Japan Science and Technology (JST), a grant from Japan Health Promotion & Fitness Foundation, TMDU priority research areas grant and National Research Institute for Earth Science and Disaster Resilience.”

Reviewers' comments:

Reviewer's Responses to Questions

**Comments to the Author**

1. Is the manuscript technically sound, and do the data support the conclusions?

Reviewer #1: Yes

Reviewer #2: Yes

2. Has the statistical analysis been performed appropriately and rigorously? 

Reviewer #1: Yes

Reviewer #2: Yes

3. Have the authors made all data underlying the findings in their manuscript fully available?

Reviewer #1: Yes

Reviewer #2: Yes

4. Is the manuscript presented in an intelligible fashion and written in standard English?

Reviewer #1: Yes

Reviewer #2: Yes

5. Review Comments to the Author

Reviewer #1: The authors have explored a highly intriguing issue, especially in today's context where aging is becoming increasingly prevalent. The study suggests that older adults who watch sports on-site experience an increase in happiness, while those who watch through television or the internet do not see a significant impact on their happiness.

I believe this article contributes an interesting perspective on how to maintain and enhance the happiness of older adults. The structure of the paper adheres to the standards of scientific research papers, with each section closely connected, clear logic, and easy to understand.

The only regret I have is that the references cited by the authors in the introduction and discussion sections only go up to the year 2022. This could lead professional peers to believe that there is a lack of literature on the topic discussed by the authors in the most recent years, 2023-2024. If possible, I recommend that the authors supplement the paper with some relevant literature from the past two years.

In the introduction section, has there been any recent attention from peers in the past two years regarding the relationship between passive sports participation and the happiness of older adults? I believe this is an area that the authors could consider expanding upon in the introduction.

In the discussion section, the authors specifically address two issues of our concern: (1) "Our findings suggest that individuals who watch sports on-site more frequently are not necessarily happy."; (2) "The relationship between watching sports and happiness was observed among older adults who did not participate in sports clubs rather than among those who participated at least several times a year." These two conclusions clearly defy readers' intuitive expectations, which is an intriguing finding. The authors have discussed these two reader concerns, but as a reader, I feel that the discussion could be more exhaustive. I hope the authors can provide more detailed explanations. If the authors could incorporate more recent literature to elucidate this phenomenon, it might be more persuasive.

Reviewer #2: Dear Authors, I hope you find in a good health. First of all I would like to congratulate for the research.

there are some comments to improve the text:

1. abstract most be crearly about the aim of the study; it seems to be different in the text;

2. in the method, were possible to have an instrumentation item and another one to describe the population;

3. in the results, there will be possible to highlight the results of the Happiness score (maybe in a separate table);

4. some suggestions must be done with the specific purpose, with the findings, suggestions to increase happiness of these specific population.

6. PLOS authors have the option to publish the peer review history of their article (what does this mean? ). If published, this will include your full peer review and any attached files.

**Do you want your identity to be public for this peer review?** For information about this choice, including consent withdrawal, please see our Privacy Policy .

Reviewer #1: No

Reviewer #2: **Yes: ** Dr. Vinicius Barroso Hirota

---

## [Author Response · Author response to Decision Letter 1]

7 Nov 2024

Dear Editor,

We sincerely thank the reviewers for their careful evaluation of our manuscript and their constructive comments. We have addressed all comments and revised our manuscript accordingly. Below are our point-by-point responses.

Response to Reviewer #1:

Comment 1: The references cited in the introduction and discussion sections only go up to the year 2022. This could lead professional peers to believe that there is a lack of literature on the topic in the most recent years, 2023-2024.

Comment 2: Has there been any recent attention from peers in the past two years regarding the relationship between passive sports participation and the happiness of older adults?

Response: We appreciate this valuable suggestion. We have conducted an additional literature review to include the most recent studies from 2023-2024 and added some new insights into online sports watching and biological mechanism.

Changes: We have updated references in our introduction. The list of new references is provided as follows:

• Yoshida M, Sato M, Doyle J. Feeling vital by watching sport: The roles of team identification and stadium attendance in enhancing subjective vitality. J Sport Manag. 2023;37: 229-242. doi: 10.1123/jsm.2021-0174.

• Kinoshita K, Matsuoka H. Risk and psychological return: a moderating role of COVID-19 risk perception in the impact of team identification on vitality after sport spectatorship in Tokyo. Int J Sports Mark Spons. 2023;24: 20-37. doi: 10.1108/IJSMS-01-2022-0007.

• Lin Y-H, Chen C-Y, Lin Y-K, Lee C-Y, Cheng C-Y. Effects of Online Video Sport Spectatorship on the Subjective Well-Being of College Students: The Moderating Effect of Sport Involvement. Int J Environ Res Public Health. 2023; 20:4381. doi: 10.3390/ijerph20054381.

• Kinoshita K, Nakagawa K, Sato S. (2024). Watching sport enhances well-being: evidence from a multi-method approach. Sport Manage Rev. 2024;27: 595-619. doi: 10.1080/14413523.2024.2329831

We have modified the introduction section with recent references:

“Previous studies have reported that passive sports participation is associated with greater subjective well-being [10-12]” (p.4, line 65).

“According to a study focusing on college students [19], online video sport spectatorship may improve subjective well-being, with sport participation moderating the relationship between online video sport spectatorship and well-being.” (p. 4, lines 78–81).

We have updated the discussion section with recent references:

“In terms of biological mechanisms, watching popular sports increases brain activity in regions associated with positive emotions, leading to enhanced self-reported well-being [34].” (p. 24, lines 300-302).

Comment 3: The discussion of two counterintuitive findings (frequency of watching and non-participation in sports clubs) could be more exhaustive.

Response: Thank you for this valuable suggestion. We appreciate the reviewer's insight that these counterintuitive findings require more detailed explanation. We have expanded our discussion of these findings with additional theoretical frameworks and supporting evidence.

Changes: We have modified the text in the discussion (p. 24–25, lines 308–322): “While frequent sports watching might intuitively seem more beneficial, moderate frequency (a few times per year) appears optimal for happiness. The complex relationship between watching frequency and happiness can be explained by three mechanisms. First, regular spectators often experience stronger negative emotions from losses, which can offset positive effects [35]. This emotional variability may explain why moderate frequency (a few times per year) shows optimal effects on happiness. Second, a study on the connection between watching sports and well-being [36] indicated that an individual’s psychological connection with a sports team may be more important than watching frequency itself for greater well-being. The relationship between watching sports and happiness may depend more on how much attachment individuals have to their favorite teams rather than how often they watch sports. Third, the hedonic adaptation theory suggests that people tend to return to a baseline level of happiness despite repeated positive experiences [37]. Frequent sports watching may lead to habituation, reducing its emotional effect over time. Thus, the relationship between sports watching and happiness appears to be influenced by both quantitative and qualitative aspects of sports engagement.”

We have updated the following text in the discussion (p. 26, lines 337–351): “The effect of watching sports on happiness may vary depending on older adults' active participation in sports. Many studies have consistently demonstrated that active participation in sports can lead to higher happiness levels [9]. However, it is generally not easy to encourage older adults, especially those with functional decline or limited mobility, to engage in regular sports or exercise. Our findings suggest that watching sports may have different implications for participants versus non-participants in sports clubs. Sports club participants may already receive social and emotional benefits through their active participation, creating a ceiling effect for additional benefits from watching sports. In contrast, non-participants may experience unique social connection and sports-related enjoyment primarily through watching sports. Additionally, the novelty effect may be stronger for non-participants. While sports club participants are regularly exposed to sports environments, individuals who do not participate in sports may find the atmosphere and excitement of live sports events more stimulating and beneficial on their emotional state. However, these potential mechanisms remain hypothetical, and future research is needed to examine the psychological processes underlying the differential effects of sports watching between sports participants and non-participants.”

Response to Reviewer #2:

Comment 1: Abstract must be clearly about the aim of the study; it seems to be different in the text.

Response: Thank you for this important observation regarding the consistency of our study aims. We have revised the abstract to ensure consistency between the stated aims in the abstract and main text.

Changes: The objective statement in the abstract (p. 2, lines 22–24) was modified to read: “This study aimed to investigate the association between different forms of sports spectatorship (on-site and on TV/Internet) and happiness among older adults in Japan.”

Comment 2: In the method, were possible to have an instrumentation item and another one to describe the population.

Response: Thank you for this constructive suggestion to improve the organization of our methods section. We have restructured the methods section to include separate subsections for instrumentation and population description.

Changes: Added two new subsections in the methods:

1. “Study population” (p. 6, lines 105-112): “Between November and December 2022,...”

2. “Measures” (p. 6–7, lines 113-160): “In our research,...”

Comment 3: In the results, there will be possible to highlight the results of the Happiness score (maybe in a separate table).

Response: Based on your suggestion, we have created a new table specifically focusing on happiness scores and their distribution across different viewing patterns.

Changes: Added S1 Table: “Distribution of happiness scores by frequency of watching sports.” (Supporting information) and the following text in the results (p.10, lines 201–208):

“S1 Table shows the distribution of happiness scores by frequency of watching sports. Participants who watched sports on-site over a few times a year reported higher happiness scores (mean = 7.6) than those who never watched sports (mean = 7.2). Regarding professional sports on-site, happiness scores were similar between those who never watched (mean = 7.6) and those who watched a few times a year (mean = 7.7). Participants who watched sports on TV/Internet at least once a month showed higher happiness scores (mean = 7.4) than those who never watched (mean = 7.1).”

S1 Table. Distribution of happiness scores by frequency of watching sports.

Frequency of watching sports Mean (SD)

Watching sports on-site

Never 7.2 (1.8)

A few times a year 7.6 (1.6)

At least once a month 7.6 (1.8)

Watching professional sports on-site

Never 7.6 (1.7)

A few times a year 7.7 (1.6)

At least once a month 7.3 (1.7)

Watching sports on TV/Internet

Never 7.1 (2.0)

A few times a year 7.2 (1.8)

At least once a month 7.4 (1.7)

SD: Standard deviation

Comment 4: Some suggestions must be done with the specific purpose, with the findings, suggestions to increase happiness of these specific population.

Response: We appreciate your suggestion regarding the need for practical implications derived from our findings. We have added specific recommendations based on our findings.

Changes: We have modified the text in the discussion (p. 27, lines 370–381): “Our research reveals a positive association between watching sports on-site and happiness, but not with TV or Internet watching. Our findings have several implications. First, it suggests the importance of creating environments that promote watching sports on-site rather than on TV or Internet. While watching sports on TV or the Internet is accessible, visiting stadiums or arenas for spectatorship may be more effective in enhancing happiness [40]. Second, optimizing the use of existing facilities, such as public parks and stadiums, could help promote happiness among older adults [46]. These venues could be designed to facilitate the social aspects of live sports events that may not be obtained through TV or Internet watching. Third, providing older adults with discounted or free tickets to sports events could serve as an effective population-based strategy for promoting happiness [18]. Overall, initiatives that encourage on-site sports spectatorship could play a key role in improving happiness among older adults.”

---

## [Decision Letter · Decision Letter 1]

6 Dec 2024

PONE-D-24-42023R1Watching sports and happiness among older adults in Japan: The JAGES cohort studyPLOS ONE

Dear Dr. Kawaguchi,

Thank you for submitting your manuscript to PLOS ONE. After careful consideration, we feel that it has merit but does not fully meet PLOS ONE’s publication criteria as it currently stands. The manuscript requires further scrutiny and additional modifications before it can be considered for final acceptance. Therefore, we invite you to submit a revised version of the manuscript that addresses the points raised during the review process.

Please submit your revised manuscript within Jan 20 2025 11:59PM. If you will need more time than this to complete your revisions, please reply to this message or contact the journal office at plosone@plos.org . Please include the following items when submitting your revised manuscript:

We look forward to receiving your revised manuscript.

Kind regards,

Mohamed Ahmed Said, Ph.D.

Academic Editor

PLOS ONE

Journal Requirements:

Reviewers' comments:

Reviewer's Responses to Questions

**Comments to the Author**

1. If the authors have adequately addressed your comments raised in a previous round of review and you feel that this manuscript is now acceptable for publication, you may indicate that here to bypass the “Comments to the Author” section, enter your conflict of interest statement in the “Confidential to Editor” section, and submit your "Accept" recommendation.

Reviewer #1: All comments have been addressed

Reviewer #2: All comments have been addressed

2. Is the manuscript technically sound, and do the data support the conclusions?

Reviewer #1: Yes

Reviewer #2: Yes

3. Has the statistical analysis been performed appropriately and rigorously? 

Reviewer #1: Yes

Reviewer #2: Yes

4. Have the authors made all data underlying the findings in their manuscript fully available?

Reviewer #1: Yes

Reviewer #2: Yes

5. Is the manuscript presented in an intelligible fashion and written in standard English?

Reviewer #1: Yes

Reviewer #2: Yes

6. Review Comments to the Author

Reviewer #1: The authors have made great efforts to actively address the suggestions for improvement put forward by me, thus enhancing the academic level of the paper.

Reviewer #2: Dear Author(s), I hope you find well! Here you can find the considerations for the manuscript to improve the publication;

Incorporation of Reviewer Feedback:

The revised manuscript includes detailed responses to the reviewers' suggestions, such as adding more recent literature (up to 2023-2024), which broadens the theoretical and practical foundation of the study.

Comments on the clarity of the study's objective in the abstract and consistency with the main text were addressed. The abstract has been revised but could still be enhanced for greater impact. Suggested improvements include:

Avoiding redundant language, such as "Although the relationship... remains unclear."

More explicitly highlighting the practical implications directly in the abstract.

The introduction mentions several theoretical references but could include a clearer explanation of why happiness in older adults is particularly relevant in the Japanese context. This would strengthen the rationale behind the study's focus.

Methodological Structuring:

The revised document organizes the methodology into clearer subsections, such as "Study Population" and "Measures," ensuring greater transparency and accessibility.

Expanded Discussion and Analyses:

The discussion in the revised manuscript presents more in-depth analyses, such as explanations for counterintuitive findings and theoretical considerations about the moderate frequency of sports watching and participation in sports clubs.

The revised text adds references to theories like hedonic adaptation and psychological impacts.

In the discussion, some ideas, such as hedonic adaptation and the psychological impact of watching sports, could be connected to other areas, such as mental health and social interaction in older adults.

Exploring why moderate frequency of sports watching appears more effective in enhancing happiness could generate additional insights.

Presentation of Results:

The revised manuscript includes supplementary detailed tables describing happiness scores concerning the frequency of sports watching, offering greater visual clarity and understanding.

Practical Applications:

New practical suggestions were incorporated to promote happiness among older adults, such as initiatives to enhance access to live sports events. The mentioned practical implications could be elaborated on, such as:

Concrete examples of public policies or initiatives to encourage older adults to attend live sports events.

Strategies to overcome specific barriers, such as financial or mobility limitations.

These changes indicate significant progress, addressing reviewers' recommendations and improving the overall quality of the manuscript. If you wish, I can delve into specific details or other aspects of the document.

7. PLOS authors have the option to publish the peer review history of their article (what does this mean? ). If published, this will include your full peer review and any attached files.

**Do you want your identity to be public for this peer review?** For information about this choice, including consent withdrawal, please see our Privacy Policy .

Reviewer #1: No

Reviewer #2: **Yes: ** Vinicius Barroso Hirota, PhD.

---

## [Author Response · Author response to Decision Letter 2]

14 Dec 2024

Dear Editor and Reviewers,

Thank you for your thoughtful and constructive feedback on our manuscript. We greatly appreciate the time and expertise you have dedicated to helping us improve our work. We have carefully considered all comments and have made substantial revisions to address each point raised. Below, we detail our responses to the reviewer's comments and the corresponding changes made to the manuscript.

Response to Reviewer #2:

Comment 1: The introduction mentions several theoretical references but could include a clearer explanation of why happiness in older adults is particularly relevant in the Japanese context.

Response: We appreciate your suggestion regarding the abstract’s clarity and impact.

Changes:

We have:

• Removed redundant language, specifically the phrase "Although the relationship... remains unclear"

• Revised the objective statement to be more direct: "This study examined associations between different forms of sports spectatorship (on-site and TV/Internet) and happiness among older adults in Japan"

• Strengthened the conclusion to better highlight practical implications: “These findings highlight the importance of developing targeted interventions that promote older adults’ access to live sports events as a public health strategy.”

Comment 2: The introduction mentions several theoretical references but could include a clearer explanation of why happiness in older adults is particularly relevant in the Japanese context.

Response: We thank you for noting the need for stronger contextualization.

Changes: We have added new text in the introduction that specifically addresses (p. 4, lines 100–101): “However, existing literature has not yet fully addressed this issue among older adults [20], which is particularly crucial in Japan, the most rapidly aging country worldwide.”

Comment 3: Exploring why moderate frequency of sports watching appears more effective in enhancing happiness could generate additional insights.

Response: We have substantially expanded the discussion section.

Changes: p. 24–25, lines 330–347: “Moderate attendance may provide a balance by allowing individuals to enjoy the excitement of live events while ensuring sufficient emotional recovery between experiences. This pattern can help minimize the accumulation of negative emotions resulting from frequent exposure to team losses or disappointments. Second, quality of engagement appears to be more critical for happiness than quantity of attendance. Evidence on the connection between watching sports and well-being [36] indicates that an individual’s psychological connection with a sports team may be more important than watching frequency itself for greater well-being. The depth of emotional attachment to a favorite team, rather than the frequency of attendance, may drive the benefits of happiness associated with sports spectatorship. Third, hedonic adaptation theory offers additional insights into why occasional attendance may yield greater happiness benefits. This theory suggests that people tend to return to a baseline level of happiness despite repeated positive experiences [37]. While occasional attendance may provide novelty, excitement, and a temporary boost in mood, frequent attendance may reduce positive emotional returns as individuals become accustomed to the experience. Less frequent attendance also enhances the psychological benefits of anticipation and savoring, further amplifying the positive emotional effects of each event. Moderate attendance may balance excitement with emotional recovery, avoiding negative emotions associated with frequent losses, or team-related disappointments.”

Comment 4: The mentioned practical implications could be elaborated on, such as: Concrete examples of public policies or initiatives to encourage older adults to attend live sports events. Strategies to overcome specific barriers, such as financial or mobility limitations.

Response: Thank you for this constructive suggestion to improve our discussion section.

Changes: Following your suggestion, we have elaborated on practical implications (p. 27–28, lines 422–437) : “Our research revealed that on-site sports spectatorship, but not TV or Internet viewing, is positively associated with happiness among older adults. While watching sports on TV or the Internet is accessible, visiting stadiums or arenas for spectatorship may be more effective in enhancing happiness [40]. Facilitating access to live sports events in older adults may function as a public health strategy to enhance happiness. This strategy could include both financial interventions, such as targeted ticket discounts or voucher programs [18], and infrastructure optimization of existing facilities, such as public stadiums and parks [46]. Stadiums and arenas could integrate age-friendly features, including ergonomic seating, accessible restrooms, and designated rest areas, to better accommodate older spectators. Hosting pre-event social gatherings or arranging guided group visits might also improve attendance and foster greater engagement among older adults. Such comprehensive approaches to promoting live sports spectatorship may serve as an effective public health strategy to enhance happiness among older adults.”

---

## [Decision Letter · Decision Letter 2]

16 Jan 2025

PONE-D-24-42023R2Watching sports and happiness among older adults in Japan: The JAGES cohort studyPLOS ONE

Dear Dr. Kawaguchi,

Thank you for submitting your manuscript to PLOS ONE. After careful consideration, we feel that it has merit but does not fully meet PLOS ONE’s publication criteria as it currently stands. Therefore, we invite you to submit a revised version of the manuscript that addresses the points raised during the review process.

We look forward to receiving your revised manuscript.

Kind regards,

Mohamed Ahmed Said, Ph.D.

Academic Editor

PLOS ONE

Journal Requirements:

Reviewers' comments:

Reviewer's Responses to Questions

**Comments to the Author**

1. If the authors have adequately addressed your comments raised in a previous round of review and you feel that this manuscript is now acceptable for publication, you may indicate that here to bypass the “Comments to the Author” section, enter your conflict of interest statement in the “Confidential to Editor” section, and submit your "Accept" recommendation.

Reviewer #1: All comments have been addressed

Reviewer #3: All comments have been addressed

2. Is the manuscript technically sound, and do the data support the conclusions?

Reviewer #1: Yes

Reviewer #3: Yes

3. Has the statistical analysis been performed appropriately and rigorously? 

Reviewer #1: Yes

Reviewer #3: Yes

4. Have the authors made all data underlying the findings in their manuscript fully available?

Reviewer #1: No

Reviewer #3: Yes

5. Is the manuscript presented in an intelligible fashion and written in standard English?

Reviewer #1: Yes

Reviewer #3: Yes

6. Review Comments to the Author

Reviewer #1: The revised manuscript has effectively addressed the reviewers' suggestions with significant revisions, and it has now reached a level suitable for publication. I recommend acceptance for publication.

Reviewer #3: The manuscript under examination explores the correlation between sports viewership and happiness in Japanese senior citizens. The work utilizes robust statistical techniques and presents valuable findings; yet certain aspects require clarification, elaboration, and methodological enhancement. The following are targeted remarks and recommendations for improving the quality and efficacy of the manuscript.

Methodology

The sampling methodology and the criteria for inclusion and exclusion require more comprehensive elucidation. Elucidate the methods employed for participant recruitment and the justification for the inclusion and exclusion criteria established. This is especially significant considering that approximately fifty percent of the original participants were eliminated, which raises issues regarding attrition bias and the generalizability of the findings.

Specify the methods employed to contact participants (e.g., via community centers, health clinics, or online platforms).

Elucidate the techniques for distributing questionnaires (e.g., postal surveys, face-to-face interviews, or electronic forms) to offer insights into response rates and any biases.

Elucidate the methodology employed to assess ADL independence. Indicate if validated instruments or standardized scales were employed and their significance to the study context.

Summarize the Japanese short version of the Geriatric Depression Scale (GDS), detailing its objective, framework, and methodology for assessing depression symptoms.

Emphasize the scale's validity and reliability to substantiate its application in the research.

Results and Statistical Analysis

The elimination of about fifty percent of the subjects substantially affects the study's generalizability. Identify potential obstacles associated with this issue and offer arguments or techniques for alleviating attrition bias.

Explore alternate metrics for happiness, such ordinal regression or continuous happiness scores, in additional analysis to maintain information integrity and evaluate the robustness of dichotomized outcomes.

Directly tackle multicollinearity concerns, especially within multivariable models.

Enhance transparency by disclosing subgroup sizes and addressing restrictions arising from small samples or diminished statistical power.

Incorporate standardized coefficients to offer detailed insights into the comparative influence of various forms of sports viewing on happiness.

Discussion

Recognize the constraints on the generalizability of findings stemming from the elevated exclusion rate and the particular demographic concentration on Japanese older persons.

Elucidate the impact of various forms of sports viewership on happiness and assess how these results correspond with or contrast to the current body of knowledge.

Elaborate on potential biases and constraints, and provide avenues for future research, such the replication of the study in other populations or the utilization of longitudinal designs.

7. PLOS authors have the option to publish the peer review history of their article (what does this mean? ). If published, this will include your full peer review and any attached files.

**Do you want your identity to be public for this peer review?** For information about this choice, including consent withdrawal, please see our Privacy Policy .

Reviewer #1: No

Reviewer #3: **Yes: ** ahmed khalifa

---

## [Author Response · Author response to Decision Letter 3]

13 Feb 2025

Dear Reviewers,

Thank you very much for your insightful and constructive feedback on our manuscript. We appreciate the time and effort you have dedicated to helping us improve our work. We have carefully considered each of your comments and have revised the manuscript accordingly. Below, we provide a detailed response to each point raised.

Methodology

#1. The sampling methodology and the criteria for inclusion and exclusion require more comprehensive elucidation.

#2. Elucidate the methods employed for participant recruitment and the justification for the inclusion and exclusion criteria established. This is especially significant considering that approximately fifty percent of the original participants were eliminated, which raises issues regarding attrition bias and the generalizability of the findings.

#3. Specify the methods employed to contact participants (e.g., via community centers, health clinics, or online platforms).

#4. Elucidate the techniques for distributing questionnaires (e.g., postal surveys, face-to-face interviews, or electronic forms) to offer insights into response rates and any biases.

Response: We sincerely thank the reviewers for their valuable feedback regarding the methodology and the potential implications of sample attrition. We have addressed these points comprehensively in the revised manuscript and provide a detailed response below. We have expanded the Methods section to provide a more thorough description of the JAGES study design. To address the potential for attrition bias, we conducted a detailed comparison of the background characteristics between our analytic sample and the original 2019 sample.

Lines 89–102:

“Data were obtained from the Japan Gerontological Evaluation Study (JAGES) [21], an ongoing cohort study examining social factors influencing the health of community-dwelling individuals aged 65 years and older. Since 2003, JAGES surveys have been conducted approximately every three years in collaboration with municipal governments across Japan [21]. These governments develop and implement long-term care insurance plans, including preventive care services. JAGES surveys provide critical data for assessing policy effectiveness and guiding future planning [21]. Within each municipality, eligible individuals aged 65 years and older without long-term care certification are selected from the resident registry. In small municipalities, all eligible older adults are included, whereas in large municipalities, 30% of the target sample is randomly selected. Among the remaining 70%, individuals aged 65 to 67 years are chosen proportionally based on population distribution. Subsequently, individuals aged 68 years and older who participated in the most recent survey are prioritized for recruitment. Additional participants aged 68 years and older are enrolled until the target sample size is achieved.”

Lines 123–126:

“The JAGES 2022 survey, conducted between November and December 2022, mailed questionnaires containing items related to sports spectatorship to 42,096 community-dwelling individuals aged ≥ 65 years, identified through official residential registers provided by municipal governments.”

#5. Elucidate the methodology employed to assess ADL independence. Indicate if validated instruments or standardized scales were employed and their significance to the study context.

Response: Thank you for raising this point. We have expanded the Methods section to further clarify that ADL independence was assessed through a self-reported question: "Do you require nursing care or assistance in daily life from anyone?"

Lines 129–130:

“Lack of independence in ADLs was defined based on self-reported responses to the question, ‘Do you require nursing care or assistance in daily life from anyone?’”

#6. Summarize the Japanese short version of the Geriatric Depression Scale (GDS), detailing its objective, framework, and methodology for assessing depression symptoms. Emphasize the scale's validity and reliability to substantiate its application in the research.

Response: Thank you for the suggestion. We have provided a more detailed description of the Japanese short version of the GDS in the Methods section. The revised text now includes evidence of its validity and reliability in assessing depressive symptoms among older adults in Japan.

Lines 172–176:

“The Japanese short version of the Geriatric Depression Scale (GDS) was used to assess depressive symptoms. This 15-item scale, validated for older adults in Japan, has demonstrated good reliability (Cronbach’s α = 0.80) and sensitivity (96%) and specificity (95%) for detecting major depressive disorder [26]. Scores range from 0 to 15, with higher scores indicating more severe depressive symptoms.”

Results and Statistical Analysis

#7. The elimination of about fifty percent of the subjects substantially affects the study's generalizability. Identify potential obstacles associated with this issue and offer arguments or techniques for alleviating attrition bias.

Response: We acknowledge this limitation. We have performed a comparison between our analytic sample and the original sample to assess potential selection bias (reported in S2 Table and discussed in the Results section).

Lines 240–244:

“To assess potential selection bias from sample attrition, we compared our analytic sample with the original sample in 2019 (n=22,528; S2 Table). The background characteristics of our analytic sample were largely comparable to those of the original sample, although our analytic samples included slightly younger participants (mean age: 73.6 vs 74.5 years).”

#8. Explore alternate metrics for happiness, such ordinal regression or continuous happiness scores, in additional analysis to maintain information integrity and evaluate the robustness of dichotomized outcomes.

Response: We appreciate this suggestion and have already performed linear regression using happiness scores as a continuous variable. The results were consistent with the dichotomized outcome analysis, ensuring the robustness of our findings. We have added this detail to the manuscript, along with a discussion of its implications. By using both continuous and dichotomized outcomes, we provide a comprehensive evaluation of the relationship between sports spectatorship and happiness.

#9. Directly tackle multicollinearity concerns, especially within multivariable models.

Response: Thank you for raising this concern. We assessed multicollinearity by calculating the Variance Inflation Factor (VIF) for each model. All VIF values were less than 10, indicating no significant multicollinearity. We have explicitly stated this in the Methods section of the revised manuscript.

Lines 212–214:

“To assess multicollinearity, we calculated the Variance Inflation Factor (VIF) for each model. All VIFs values were less than 10, indicating no significant multicollinearity in the models.”

#10. Enhance transparency by disclosing subgroup sizes and addressing restrictions arising from small samples or diminished statistical power.

Response: We have ensured that the sizes of all subgroups are clearly reported in the Results section and relevant tables.

Lines 314–317, 322, 327

“In the subgroup analysis of participation in sports clubs at least a few times a year (n = 4,091), none of the three forms of watching sports was associated with a higher happiness score (as presented in Table 5). However, the subgroup analysis of non-participation in sports clubs (n = 7,174)”

“In the sex-stratified analysis (male: n = 5,382; female: n = 5,883),”

“In the age-stratified analysis (< 75 years: n = 6,653; ≥ 75 years: n = 4,612),”

#11. Incorporate standardized coefficients to offer detailed insights into the comparative influence of various forms of sports viewing on happiness.

Response: We appreciate your suggestion. We have reported both unstandardized and standardized coefficients.

Discussion

#12. Elucidate the impact of various forms of sports viewership on happiness and assess how these results correspond with or contrast to the current body of knowledge.

Response: We appreciate the reviewer's insightful suggestion to further elucidate the differential impact of various forms of sports viewership on happiness. We have revised the Discussion section to explicitly highlight the contrasting of on-site versus TV/Internet spectatorship. Specifically, we have modified the text to emphasize that the observed positive association between watching sports and happiness was unique to on-site spectatorship and was not found for TV/Internet viewing.

Lines 406–407

“The positive association between watching sports and happiness was specific to on-site spectatorship and not observed with TV/Internet watching.”

#13. Recognize the constraints on the generalizability of findings stemming from the elevated exclusion rate and the particular demographic concentration on Japanese older persons.

#14. Elaborate on potential biases and constraints, and provide avenues for future research, such the replication of the study in other populations or the utilization of longitudinal designs.

Response: Thank you for your suggestion. We have explicitly acknowledged these limitations in the revised manuscript and expanded the Limitations section.

Lines 474–491

“This study has several limitations. First, generalizability is limited by our focus on community-dwelling older adults in Japan. Cultural factors may influence the relationship between sports spectatorship and happiness, warranting caution when extending these findings to other populations. Second, while our analytic sample was largely comparable to the original sample, the sample attrition could have introduced selection bias, potentially skewing the sample towards more active individuals. Additionally, although we adjusted for several confounders, unmeasured variables such as sporting event type, motivation for attendance, and team preferences were not considered due to questionnaire constraints [47]. The COVID-19 pandemic may also have affected our data, as concerns about infection could have diminished enjoyment and attenuated feelings of happiness among spectators [48]. Furthermore, the primarily cross-sectional design restricts our ability to infer causality. Although we attempted to reduce reverse causation by using covariates measured prior to exposure, we cannot eliminate the possibility that happier individuals are simply more inclined to watch sports. Future research should address these limitations by replicating this study in diverse populations across different cultures. Longitudinal designs are needed to establish temporal relationships and elucidate the dynamic interplay between sports spectatorship and happiness. Further investigation into underlying mechanisms (e.g., social interaction, emotional responses, team identification) and the moderating effects of sport type and viewing context is also warranted.”

---

## [Decision Letter · Decision Letter 3]

16 Feb 2025

Watching sports and happiness among older adults in Japan: The JAGES cohort study

PONE-D-24-42023R3

Dear Dr. Kenjiro Kawaguchi,

We’re pleased to inform you that your manuscript has been judged scientifically suitable for publication and will be formally accepted for publication once it meets all outstanding technical requirements.

Kind regards,

Mohamed Ahmed Said, Ph.D.

Academic Editor

PLOS ONE

Additional Editor Comments (optional):

Reviewers' comments:

Reviewer's Responses to Questions

**Comments to the Author**

1. If the authors have adequately addressed your comments raised in a previous round of review and you feel that this manuscript is now acceptable for publication, you may indicate that here to bypass the “Comments to the Author” section, enter your conflict of interest statement in the “Confidential to Editor” section, and submit your "Accept" recommendation.

Reviewer #3: All comments have been addressed

2. Is the manuscript technically sound, and do the data support the conclusions?

Reviewer #3: Yes

3. Has the statistical analysis been performed appropriately and rigorously? 

Reviewer #3: Yes

4. Have the authors made all data underlying the findings in their manuscript fully available?

Reviewer #3: Yes

5. Is the manuscript presented in an intelligible fashion and written in standard English?

Reviewer #3: Yes

6. Review Comments to the Author

Reviewer #3: The manuscript under review examines the relationship between sports viewing and happiness in Japanese elderly. The work uses powerful statistical techniques and provides valuable results; no further modifications are needed.

7. PLOS authors have the option to publish the peer review history of their article (what does this mean? ). If published, this will include your full peer review and any attached files.

**Do you want your identity to be public for this peer review?** For information about this choice, including consent withdrawal, please see our Privacy Policy .

Reviewer #3: No

---

## [Editor Report · Acceptance letter]

PONE-D-24-42023R3

PLOS ONE

Dear Dr. Kawaguchi,

I'm pleased to inform you that your manuscript has been deemed suitable for publication in PLOS ONE. Congratulations! Your manuscript is now being handed over to our production team.

Kind regards,

on behalf of

Dr. Mohamed Ahmed Said

Academic Editor

PLOS ONE